# The Impacts of Traditional Culture on Small Industries Longevity and Sustainability: A Case on Sundanese in Indonesia

Anne Charina [1,*][ID], Ganjar Kurnia [1], Asep Mulyana [2][ID] and Kosuke Mizuno [3]

1   Faculty of Agriculture, Universitas Padjadjaran, Sumedang 45363, Indonesia
2   Faculty of Economics and Business, Universitas Padjadjaran, Sumedang 45363, Indonesia
3   Development Studies, Kyoto University, Kyoto 604-8103, Japan
*   Correspondence: anne.charina@unpad.ac.id

**Abstract:** This study investigates traditional culture as one of the factors of the longevity and cross-generation sustainability of Sundanese small industries in Indonesia. The failure rate of small industries in Indonesia is high, and thus, this study is critical. We mapped the relationship between Hofstede's cultural dimensions, longevity, and business sustainability in ten selected Sundanese small industries surviving up to three generations. Data were obtained from semi-structured interviews and company data. The results revealed that the strength of Sundanese cultural traditions, including high long-term orientation, high collectivism, low power distance, and high indulgences, positively contribute to the longevity of small industries in Indonesia. In addition, the social performance of Sundanese entrepreneurs is mainly based on their religious values and a highly collectivist culture; educational experience also affects their environmental performance. This study highlights the need to understand the traditional culture, which can play an essential role in achieving business longevity but also can present some limitations, especially in terms of economic performance. Therefore, to create a sustainable small industry, efforts are needed to change the mindset of Sundanese entrepreneurs to be more open to an innovative global culture while maintaining local values that positively contribute to business.

**Keywords:** small industry; culture; Hofstede; Sundanese; longevity; sustainability

## 1. Introduction

Indonesia is a developing country dominated by small industries (62.9 million businesses or 99.99% of the total existing businesses). Small industries are critical, and make up 97% of Indonesia's workforce [1]. Despite the great potential of small industries, many cannot survive due to various limitations, including finance, manpower, infrastructure, networks, access, and other factors [2–5]. Furthermore, other research also revealed that many small industries were destroyed when they underwent a cross-generational business transformation process. Some revealed that about 70% of family-owned businesses went bankrupt after the founder retired, 17% could survive into the second generation, and only 13% could continue their business into the third generation [6]. The life span of small industries is relatively short [7] and they are considered vulnerable, with a life expectancy of only 5–10 years [2].

A previous study in East Priangan, Indonesia, investigated 360 small-scale agriculture-based industries (the most dominant industrial sub-sector in Indonesia) and found that 76% of small industries have only been run by the first generation, 20% have been run by the second generation, and only 4% managed to enter the third generation [8]. This finding implies that a cross-generational business transformation process is crucial in developing small agriculture-based industries in Indonesia. The third generation is also considered the

most critical point in cross-generational business transformation [9]. Therefore, longevity is an important topic to study in the development of small industries.

Interestingly, in Indonesia, several small industries that survive until the third generation are owned by Sundanese entrepreneurs [10]. Their businesses are related to traditional Sundanese food and handicraft products. Sundanese is a native tribe from the western part of the island of Java. Sunda is the forerunner in establishing civilization in Indonesia. The Sundanese are rich in cultural traditions passed across generations, including entrepreneurship. Among the cultures are the collectivist culture with fellow Sundanese, "gotong royong" which means collaboration, and the principle of "ngeureuyeuh asal mayeng" in running a business, which means "slowly but lasting a long time". This longevity phenomenon may occur because of the strength of the Sundanese tradition through various values applied by Sundanese entrepreneurs across generations in their business activities. Success stories of businesses in other countries surviving across generations because of the strength of their traditions were also reported in previous studies. For example, the strength of tradition and innovation are survival strategies for family businesses in Japan [11], local traditions and succession planning are the key to the survival of small businesses in Italy [12], and some Chinese culture and traditions, such as guanxi, play an important role in entrepreneurship [13]. Cultural traditions play a role in supporting business innovation in Thailand [14].

This research adopted Hofstede's cultural theory. It is considered the most relevant to show the effect of community culture on the values of its members, including how these values relate to the entrepreneurial behavior of certain ethnic groups [15]. Several previous studies have indicated how national culture affects organizational activities [16,17], and some have also examined how culture affects innovation activities [14,18], and some have conducted cross-cultural research [6,19]. The main objective of this study is to analyze the relationship between cultural dimensions, longevity, and sustainability. Therefore, our research questions are (1) What is the impact of traditional Sundanese culture on the longevity of small industries?; and (2) What is the role of Sundanese traditional culture on the performance of small industries in achieving sustainability? Considering the small number of small industries that can survive for up to three generations in Indonesia, a qualitative research approach was chosen. We believe that today's small industries are not only required to last but also endeavor to achieve business sustainability because longevity does not necessarily guarantee good economic, social and environmental performance. Currently, sustainable entrepreneurship is paramount because it provides greater benefits for the company, the community, and the environment [20,21]. We believe that understanding the characteristics, traditions, and uniqueness of Sundanese culture and learning about the limitations of small industries belonging to the Sundanese will help them achieve business sustainability. This study was structured by describing our introduction and research approach, followed by a literature review and the methodology used to obtain and analyze the data. Next, our research findings are presented with discussions, limitations, future research agendas and conlusion.

## 2. Literature Review

### 2.1. Hofstede's Cultural Theory

The definition of cultures varies; Hofstede defines culture as "the collective programming of the mind that distinguishes members of one group or category of people from others". In general, the term culture is used for tribes or ethnic groups (in anthropology), for nations (in political science, sociology and management), and for organizations (in sociology and management) [15]. Studies on the related topic mostly use Hofstede's theory. In his research, he identified six characteristic models to measure a culture in a cross-country society. The cultural dimension represents an independent preference for one state over another that distinguishes states (not individuals) from one another. Hofstede developed the original model, which emerged from factor analysis to examine the results of IBM's worldwide employee value survey between 1967 and 1973, and it has been refined since

then. The original theory proposed four dimensions in which cultural values could be analyzed: individualism-collectivism; uncertainty avoidance; power distance (strength of social hierarchy), and masculinity-femininity (task orientation versus people orientation). Independent research in Hong Kong led Hofstede to add a fifth dimension, namely long-term orientation, to include aspects of values that were not addressed in the original paradigm. Later in 2010, Hofstede added a sixth dimension, indulgence versus self-control.

Long Term versus Short Term Orientation, related to the choice of focus for people's efforts: the future or the present and past; Power Distance, related to the different solutions to the basic problem of human inequality; Individualism versus Collectivism, related to the integration of individuals into primary groups; Uncertainty Avoidance, related to the level of stress in a society in the face of an unknown future; Masculinity versus Femininity, related to the division of emotional roles between women and men; Indulgence versus Restraint, related to the gratification versus control of basic human desires related to enjoying life [15].

Research in Turkey using Hofstede's cultural theory showed that the concept of entrepreneurship varies between countries, and an entrepreneur reflects the dominant values of his national culture. Turkish culture, characterized by high collectivism, high uncertainty avoidance, and high-power distance, is associated with business [22]. On the other hand, American cultures, dominated by individualism, low power distance, and high masculinity, create many entrepreneurs. Another case is Saudi Arabia, whose culture exemplifies low individualism and high-power distance. The different cultures enrich the practical results, and the uniqueness of each culture supports the success of entrepreneurs worldwide. Our research analyzed the dimensions of Hofstede's culture in developing entrepreneurship in developing countries, such as Indonesia, especially its relation to longevity and sustainability in small industries. So far, limited studies have examined how this dimension of Hofstede's theory is attached to small entrepreneurs in Indonesia. There are many structures and contextual elements, both internal and external, that will affect the process that must be considered here. It is important to realize that some of these cultural, structural, and contextual elements overlap and reinforce each other in business activities, which requires attention.

### 2.2. Longevity and Sustainable Entrepreneurship

The concept of longevity is critical to study because small industries are vulnerable to failure. Why some companies can survive across generations while others fail is a fundamental puzzle for some research across various disciplines. To date there is no single explanation for longevity, or even a consensus on why companies should live long and survive across generations. Even though business historians tend to view survival as a measure of performance, some organizational experts assume that the firm is, at a theoretical level, an immutable entity that should at least be permanent, and is motivated by the need to survive over time [11].

We argue that the antecedent factors of longevity in small industries come from values or habits that have become traditions applied through the behavior of entrepreneurs in their business activities and then translated deeply into their internal capacities. Furthermore, there are several other external influences such as markets, competition, regulation and social capital that affect the activities of small industries. In order to expand the limited literature discussing the behaviour of small industries, we believe that exploratory studies are needed and more effectively understand the behaviour of small industries to survive across generations.

More importantly, small industries cannot rely solely on longevity to realize good performance because it is not a guarantee that the business will perform well; many other factors contribute. Previous research has revealed factors that may explain longevity, such as industry branch, company size, maturity stage, privileges, transformation ability, cultural context, historical time, and or even pure luck [9]. Therefore, sustainable entrepreneurship is currently considered essential, a condition reflecting more measurable benefits for

companies, communities, and the environment. Although it sounds ironic for small-scale industries, it is important for small industries to perform better. In the global context, sustainable entrepreneurship is defined as "the discovery and exploitation of economic opportunities through the generation of market imbalances that initiate the transformation of the sector towards a more environmentally and socially sustainable country" [23]. It combines opportunities and the intention to simultaneously create value from the economy, society, and the environment [24]. It includes economic and non-economic benefits for individuals, the economy, and society [25]. The literature defines sustainable entrepreneurship as a "purposeful teleological process" to achieve sustainable development by discovering, evaluating and exploiting opportunities and creating values, resulting in economic prosperity, social cohesion, and environmental protection [26].

We believe that small industries should not only be able to survive across generations but must also have good economic, social, and environmental performance. These need to be prepared because the process is certainly not easy, especially for small industries. So far, there is no standard measure of sustainability for small industries. Enderle [27] argued that overall, the current sustainability reporting standards commonly used by large industries might not be suitable for small industries because they were developed with general reference to large businesses. Some experts then used a variety of simpler attributes considered suitable for measuring sustainability in small industries. Economic performance is measured by the attributes of sales growth, market share, cost reduction, and profitability. Social performance is measured by the attributes of health standards for workers, employee satisfaction, customer satisfaction, relations with stakeholders, and community involvement in the business. Meanwhile, environmental performance is measured by product safety and health attributes, waste management, and factory safety and health [28–35]. Figure 1 presents our proposition to explore the effects of traditional culture on the longevity and sustainability of Sundanese businesses.

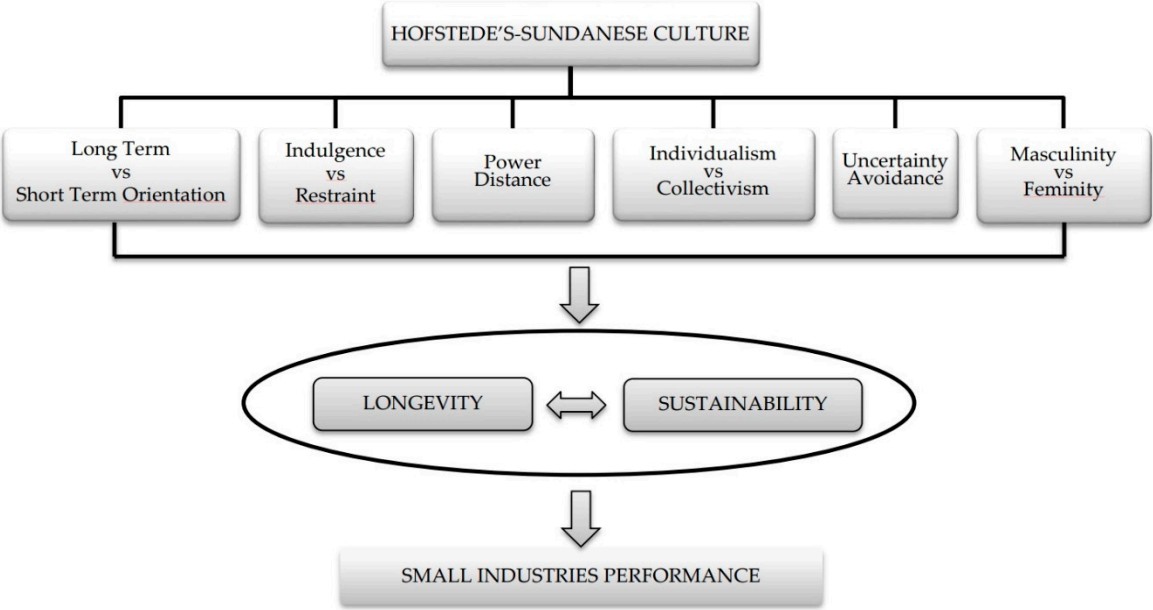

**Figure 1.** Research Model.

## 3. Materials and Methods

### 3.1. Methods and Data Collection

This study employed a qualitative method to obtain more in-depth information about the problem and provide a detailed description of complex phenomena [36]. The samples were selected by the purposive sampling technique, namely criteria sampling, as we needed the samples consisting of strategically selected objects. We have created the following

criteria to ensure that the companies are relevant. First, the company is a small industry owned and operated by Sundanese people. Based on the Regulation of the Minister of Industry of the Republic of Indonesia Number 64 of 2016, small industries are defined as industries with a maximum of 19 employees with an investment value of fewer than one billion rupiahs. Second, the company has been run for at least three generations and was still operating when this research was conducted. The geographical area studies were limited to East Priangan, West Java, Indonesia, considering that the area (Ciamis Regency, Garut Regency, Tasikmalaya Regency, Banjar, Sumedang Regency) are mainly inhabited by the Sundanese. Hence, it is considered adequately homogeneous from the point of view of business culture and industrial development paths.

Only ten small industries met the criteria for this study. These small samples are suitable for a qualitative approach, as the results will be more in-depth [37]. The ten selected small industries were industries producing traditional Sundanese food and Sundanese woven handicrafts. Each of these small industries only produces one product in their factory, except that handycraft industry, produces various handycraft products (Appendix A).

The data sources were primary and secondary. Primary data were obtained through semi-structured interviews using an interview guide developed based on the research questions included in this study: "the impact of traditional Sundanese culture on the longevity and sustainability in small industries". We did not intend to prove or disprove any particular hypothesis, but rather our focus was on the "how" and "why" of certain cultural dimensions affecting the "longevity" and "sustainability" of small industries. We interviewed the owners of the companies from December 2019 to February 2020; the duration of interviews with each company was about 60–90 min. The interviews were recorded with the consent of the interviewee. The objective performance was supported by various secondary sources from the ten small industries, such as sales reports and financial statements. This data also supported the assessment of the business performance of the ten Sundanese small industries.

### 3.2. Data Analysis

Data analysis was done in the following stages. The categories were determined based on the variables explored. The transcription was read in detail and separated into the appropriate categories, predetermined to deeply examine each category for its relationships and interrelationships [38]. To summarize the findings, the data in each category were analyzed systematically and interpreted based on patterns revealing similarities and differences in meaning, action, and/or context of the empirical data to produce in-depth findings. Several important statements of direct quotations by participants during the interview have been included to support our argument so that the reader can get an idea of the original text [39]. Company document analysis was considered for data triangulation and construct validity [40,41].

### 4. Results

### 4.1. The Impact of Sundanese Traditional Culture in Realizing the Longevity of Small Industries

This study reveals similar antecedent factors, the key determinant influencing Sundanese small industries to survive across generations. The Sundanese businessmen adhere to their traditional culture, which is the key to their business survival for three generations. The Sundanese cultural traditions are passed across generations. Sundanese have soft culture character, which is smooth language, politeness, friendliness, sociability, and attachment to local values. This local cultural value is reflected through the advice or proverbs applied and taught by parents to their children from an early age. The advice contains recommendations and prohibitions in carrying out life so that always be on the right path. These values are inherent in the Sundanese and are reflected in their life activities, including entrepreneurship.

The most prominent traditonal culture, highly reflected in Sundanese entrepreneurs, is the long-term orientation. In running their business activities, Sundanese entrepreneurs

rely on some cultural values, including "*ngeureuyeuh asal mayeng*" which implies "slowly but lasts longer", the principle of "*alon-alon asal kalakon*" which implies "slowly but surely", "*Lamun keyeng tangtu pareng*" which implies "if we keep trying our goals will surely be achieved" and many other values that imply a long-term orientation. Even the Sundanese people adhere to the principle that all wealth owned should be enjoyed up to the seventh generation. These values led to Sundanese entrepreneurs running their business activitiesenjoyed, not forcing themselves beyond their capability, not easily giving up, and having long-term principles (Table 1).

**Table 1.** Direct Quotes for "High long-term Orientation" as a factor.

| | |
|---|---|
| Company 10 | "We are Sundanese who hold the cultural value of "*ngeureuyeuh asal mayeng*" which means slow but sustainable, this makes us patient in doing business, not giving up easily." |
| Company 1 | "Although our family business journey is hard, but we adhere to the principles of our culture as Sundanese that we apply in business, including the culture of "*ngeureuyeuh asal mayeng*" ingrained in our family. Hence, when we fall, we do not lose hope; we keep being enthusiastic because business is not only about big profits. Even though our profits are not much, most importantly, we can survive." |
| Company 4 | "When I run a business, I adhere to the principle of "*alon-alon asal kalakon*" which means slowly but surely. I run the business happily instead of a burden, not forcing myself beyond my capability and doing it as it is." |
| Company 6 | "I adhere to the principle of "*Lamun keyeng tangtu pareng*" which means "if we keep trying, we will achieve our goals". I apply this in running my family business from the beginning. If we are serious about maintaining our family business, we will definitely obtain maximum results. Don't be discouraged because failure in business is common, but we should lose our motivation." |
| Company 2 | "We must remember, that the family business is a provision for our children and grandchildren not just for now. Hence, we take care of this business; the ups and downs of business are normal, but we always rise again for the sake of our children and grandchildren." |

The interviews reveal that all Sundanese entrepreneurs analyzed in this study agreed that the principles of Sundanese cultural values, especially the long-term oriented principles applied in running a business, play an important role in realizing a business that lasts across generations. Sundanese cultural values embodied in advice are numerous, reaching hundreds of wisdom. Sundanese ancestors used to educate children by passing these tips to ensure the child remains on the straight path in life. Many of these Sundanese values can be applied to the business. They are imprinted in the souls of entrepreneurs and reflected in their activities when running a business, helping them act and interact with stakeholders and to make decisions. When their businesses faltered, they could survive and never give up easily. Thus, the family business survives across generations.

Low power distance is another dimension of Sundanese culture affecting their business's longevity. It may sound ironic, but the small scale of their business enables them to be independent in running their business instead of depending on the government or any other institution. Since its establishment, their business has been very independent, without government intervention, so they are free from various rules binding them. On the other hand, the culture of low power distance among the Sundanese is also evident from the absence of power distance between business owners and their employees. Their relationships are very close, almost like siblings, so there is minimal conflict. This close relationship is the key to the sustainability of their business (Table 2).

**Table 2.** Direct Quotes for "the low Power Distance" as a factor.

| | |
|---|---|
| Company 2 | "Even though our business is small, we don't depend on anyone, including the government, we can survive without having to ask or rely on government assistance." |
| Company 4 | "When big businesses get access due to their connection to the officials, our business grows because of our independence, my grandfather and parents fought hard with their abilities to continue to maintain this business." |
| Company 7 | "Entrepreneurial activities are truly free activities, outside of government intervention, we can be creative and independent, because the nature of entrepreneurs is actually a need for independence, in order to be creative." |
| Company 8 | "Without intending to minimize the role of the government, we entrepreneurs do need freedom to work, we need independence to survive, we Sundanese are taught to be grateful for what we have, without having to expect too much from others." |
| Company 9 | "The government allows freedom to entrepreneurs, so that we can be independent, by not being tied to anyone, our space is wide, we can be creative, we can survive because we are not bound by various rules that bind us. What is important is that we do business properly and do not harm others." |
| Company 3 | "Interestingly our relationship with employees is very close. My employees often cash in first if there is a need, that's not a problem for us. Our relationship is close like brothers, so they are loyal to us to the point that many of them have worked at our factory for more than 30 years." |
| Company 5 | "There is no distance between my employees and me, we are friends and respect each other, this is important to us, we are far from conflict." |

The freedom given to small industries by the Indonesian government, by not binding them or providing burdensome regulations, is one of the factors supporting the Sundanese small industries' survival. This proves that freedom (by staying on the right track) is neededfor long lasting business. In addition, the egalitarian relationship applied by the entrepreneurs creates a more conducive work climate in the company. It minimizes conflict, an important foundation for small businesses to last a long time.

Another strong Sundanese culture embedded in the activities of the Sundanese businessman is the highly collectivistic relationship. Sundanese ethnicity is known as the "*ngariung*" culture or always gathers, including in business, they run a family business together, their family ties are very strong. Because of the strength of the collectivity relationship, the Sundanese have high social and emotional wealth (SEW), and they feel they belong to each other; the commitment to their ancestors is very strong, including in preserving their family business. They proudly uphold the good name of the family. The values of togetherness of Sundanese entrepreneurs with their fellow ethnic groups are closely reflected, and they always involve relatives and neighbors around them in their business activities.

The principle of "*gotong royong*" and mutual assistance is reflected in the recruitment of the Sundanese workforce, the majority of which are their underprivileged relatives or neighbors (Table 3). This culture of mutual assistance is believed by Sundanese entrepreneurs as the first step to achieve mutual benefit, which in the end will affect the longevity of their business.

The other dimension of Sundanese culture which plays a very important role in supporting the business longevity is the high level of indulgence of Sundanese in running a business. Indulgence in the Sundanese manifests as gratitude for all the results God gives. This principle is more driven by the religious values of Sundanese entrepreneurs. They believe that always being grateful for what is given by God makes them happy in running a business (Table 4).

**Table 3.** Direct Quotes for "the high Relationship of Collectivity" as a factor.

| | |
|---|---|
| Company 3 | "We Sundanese prioritize family commitment, we have great socio emotional wealth in this family business, this business must be maintained, my sister and I were able to pursue higher education and were raised by our parents from this business." |
| Company 9 | "The togetherness in our family is very strong, this business means a lot to our family, my brother and I work together to fight to maintain this business." |
| Company 2 | "We Sundanese strongly adhere to the principle of togetherness or collectivity, my business is dominated by all of the workforce, which are my brothers and neighbors who cannot afford it, by opening jobs for them this is one of the things that makes our business sustainable." |
| Company 4 | "The culture of collectivity or "*gotong royong*" has become the hallmark of the Sundanese people, we as business owners are compact with the workers, there is no gap between us, so that our business has minimal internal conflicts, we can last a long time because of the support from our employees, suppliers and consumers." |
| Company 1 | "One of the principles of Sundanese values is "togetherness will make us live longer" it turns out that the saying is true, with togetherness between brothers, employees, consumers, we become more confident, our business becomes more blessed, the point is maybe our business is not big profit, but we live it wholeheartedly because we are always together so that our business can last more than 60 years." |
| Company 10 | "Religion also teaches that the key to lasting life is to help each other, work together and give many benefits to the people around us first. I apply this principle in running my business, basically I hope that my business will benefit the people around us, because their happiness will be a good prayer for us to be able to survive." |

**Table 4.** Direct Quotes for "the high level of Indulgence" as a factor.

| | |
|---|---|
| Company 7 | "My parents always teach me to be grateful for what I have, including this business, I am always grateful, this is very important, so I am happy to run the business." |
| Company 8 | "The Sundanese people highly value gratitude, we Muslims are taught to be grateful for what we have." |
| Company 3 | "In my opinion, their ups and downs could be because they are not grateful for the blessings given by God, our family is very grateful for the business we have, even though it is a small business, we believe the more grateful we are, the more blessings God has given us." |
| Company 4 | "I am very grateful for our family business, we run this business happily, If we are happy, not forcing beyond our capability, and not putting too much pressure on us and are always grateful, surely our business will survive." |
| Company 5 | "Happiness is the key in life, as well as in business. We, Sundanese, value happiness in many ways; we run this business with lots of smiles and gratitude." |

Most participants agreed that happiness is the main key for them to be able to run their business for a long time. Happiness makes them enthusiastic about running a business, encourages them to persevere and means that they are always grateful for what they have achieved. This attitude plays a big role in realizing longevity.

*4.2. The Role of Sundanese Traditional Culture on Small Industry Performance in Achieving Sustainability*

Sundanese entrepreneurs have survived across generations, partly because of the strength of traditional culture and the principles of life.. The longevity of a company does not guarantee that the business has good economic, social and environmental performance. Therefore, this study also considers the relationship between the dimensions of

traditional Sundanese culture and business sustainability. We argue that the cultural traditions reflected in the values of entrepreneurs will be translated into their business culture, contributing to their business growth capacity. However, in facing globalization, small industries cannot only be satisfied with longevity, but it would be better if its supported by good economic, social and environmental performance. Thus, "business sustainability" is achieved, which will certainly benefit the company, society, and the environment.

This research investigated the current company's economic, social and environmental performance. The interviews with the company's owners and the limited sources of the company's financial statements revealed that during the last three years, the company had experienced problems in economic performance. Due to the limited data owned by the company, the economic performance of 10 small industries of Sundanese only focuses on using sales growth as an attribute. Their previous sales growth often fluctuated, but in the last three years showing a declining trend (Table 5).

**Table 5.** Sales Growth in Ten Sundanese Small Industries for 2017–2019.

| Economic Performance Attribute | Year | Company | | | | | | | | | |
|---|---|---|---|---|---|---|---|---|---|---|---|
| | | 1 | 2 | 3 | 4 | 5 | 6 | 7 | 8 | 9 | 10 |
| Sales Growth | 2017 | 33% | −20% | −27% | 6.7% | −2.5% | −2.7% | 13.6% | −3.1% | −4.4% | 8.9% |
| | 2018 | 2.5% | −12% | −19.5% | 5% | −7.6% | −8.5% | 6.5% | −9.5% | −11.5% | 4.5% |
| | 2019 | −4% | −14% | −20.1% | −11.9% | −11.1% | −12.2% | −10.4% | −14.8% | −19.3% | −10.7% |

Source: Sales Report of the Ten Companies (2017–2019).

Sales growth is a change in sales increase or decrease across years, as indicated by the company's sales and profit and loss reports. A good company can be identified from its continuing sales year by year, which affects company profits and internal funding. However, this condition is not found in the ten small industries run by the Sundanese, assuring the last three years, their product sales growth has decreased. They are used to dealing with fluctuating product sales, but this of course should not be allowed because it will endanger the company's existence.

The research results showed that one of the critical factors contributing to their weak economic performance was the lack of innovation in their business. The businesses have not experienced significant development in product quality, production process, packaging, and marketing. Poor innovation leads to monotonous and outdated product development, and this has a direct impact on market response, which in turn affects sales growth and other economic growth variables. In addition, there are other factors that may also affect sales, such as the level of competition, which is difficult to avoid. The low level of innovation in Sundanese entrepreneurs is allegedly born from the Sundanese culture, which is high in uncertainty avoidance, which means that their level of risk-taking is low, they prefer comfort zones and avoid new things that they feel are uncertain results (Table 6).

**Table 6.** Direct Quotes for "high in Uncertainty Avoidance" as a factor.

| | |
|---|---|
| Company 3 | "We, Sundanese, don't like new things that are uncertain, in running our business we choose certain things." |
| Company 9 | "We do not dare to try new things. We seem to need to learn it, we are used to something moderate." |
| Company 5 | "Trying new things in business is not easy, we have been used to doing this routine for decades, but one day our business must change, so that it doesn't stagnate." |
| Company 7 | "Sundanese people enjoy the safe zone too much, so it is difficult to start new things, it needs gradual improvement." |
| Company 1 | "Maybe we need to prepare in advance the facilities and infrastructure needed for us to innovate, we think it will take time and we have to improve for that." |
| Company 2 | "The Sundanese tradition does not like to take risks; Sundanese are too careful, We and our children need to learn this." |

All participant has realized that the low growth of their business was partly due to their culture which was always love the comfort zone, quickly satisfied with the results they received, reluctant to try new things, did not dare to take risks, so that their products became less innovative and difficult to develop in the market.

Another factor that leads to the growth of small industries owned by ethnic Sundanese entrepreneurs that is not optimal is that the owner is still identified with the business. The culture of the owner is exactly the same as the culture of the business. One of the dominant entrepreneurial powers is triggered by the tradition of high masculinity in the Sundanese ethnicity. In the Sundanese tribe, manshave a stronger position than womans; this is because women are considered to have many limitations, and there are many values that must be maintained by woman, Sundanese women are more required to stay at home. Furthermore, if she gets married, she will leave her house to follow her husband. Therefore, the role of men of Sundanese ethnicity is very high, especially in terms of achievement, control, and power.

Masculine culture tends to be male dominated, with male Sundanese characters being described as more assertive, tough, and ambitious. The succession process for many ethnic Sundanese entrepreneurs is to pass things down to the first son. It is different if the family does not have a son. All participants in this study who received the business transformation were male. The dominance of men, especially the first son in the Sundanese tribe, makes their ego high. For example, organizationally, companies have grown large enough to, in many cases, require functional managers to take over certain tasks performed by their owners. Managers need to be competent but need not be of the highest calibre, as their upward potential is limited by company goals. However, this becomes difficult, because Sundanese entrepreneurs tend to be unwilling to hire managers for their businesses, as they feel the owner is the best manager for their business. In fact, they really need skilled people to improve their economic performance, in terms of improving product growth, managing finances, and designing market expansion. However, they are not interested in global culture, due to the fear of the negative impacts of foreign cultures threatening the Sundanese culture. They are also reluctant to join community/associations or business networks because the benefits are unclear (Table 7). Whereas strategic networks and alliances will be beneficial in building collaboration and solving the company's economic problems [42,43].

**Table 7.** Direct Quotes for "high Masculinity" as a factor.

| | |
|---|---|
| Company 4 | "As the eldest son, I am very responsible for my younger siblings, my heart is united with my family business, therefore I am not interested in recruiting other people to become managers in our company." |
| Company 3 | "I have many limitations, but I feel I can still lead this business, so I'm not interested in hiring people to be managers in my business." |
| Company 6 | "I was educated by my parents to be versatile, including in leading this business, One day maybe I will hire a professional manager for this business, but not now, because I am still young and still able to lead my family business." |
| Company 7 | "The role of men in the Sundanese ethnicity is indeed big, sometimes I admit it makes me individualistic, we find it difficult to accept new people in our business, we often do not trust people outside our environment. In the future, we really have to improve." |
| Company 8 | "I am not sure about the benefits of joining a network or business association. So far, my family business has not joined any network, nor do I have any information related to business networks." |
| Company 10 | "Without eliminating the role of females, in Sundanese culture, the female's main task is taking care of the household. They rarely engage in business; they are busy with children's affairs." |

On the other hand, concerning the social and environmental performance, ten Sundanese small businesses have implemented social motives in running their businesses. The interviews showed that the social motives of small industries are mostly driven by the normative aspect, that being the religious values of the owners. Most Sundanese entrepreneurs, who are Muslim, apply religious principles in their business, for example by giving 2.5% of the profits to the poor, as described in the Sunnah of the Prophet Muhammad PBUH. This is applied across generations by entrepreneurs from the first to the third generation. In addition, the traditional culture, and highly collectivist relationships, also enhance their social performance. They involve the local community in their business by recruiting them to become employees in their company (Table 8).

**Table 8.** Direct Quotes for "Small Industries Social Initiative".

| Company 2 | "Islam encourages us to share and do social activities, including business. We are used to sharing 2.5% of the profit with the poor." |
|---|---|
| Company 4 | "It is our obligation as Muslims to help others by sharing. Every month I share 2.5% of business profits with the poor, especially those around us." |
| Company 5 | "I recruited poor neighbors to work in my company; this helped them amidst the difficulty of finding work today." |

Concerning environmental performance, the interviews revealed that environmental initiatives undertaken by 10 Sundanese small industries are mainly based on the internalization of knowledge and insights from the educational background of the company owners (Table 9). Environmentally friendly activities undertaken by Sundanese entrepreneurs include waste management, using environmentally friendly materials for food product packaging, using wood briquette fuel in the production process, and using environmentally friendly natural dyes for various handicrafts. The experience of company owners during school significantly affects their environmental interests. The eco-friendly movement has recently become a new trend in our society, especially among educated people. The educational background of entrepreneurs, especially those in the third generation, the majority of whom have a Diploma and Bachelor's degree, increase their knowledge of the importance of environmentally friendly behavior in business.

**Table 9.** Direct Quotes for "Small Industries Environmental Initiative".

| Company 1 | "I am concerned about the current state of natural pollution. At university, I have learned about the importance of environmentally friendly actions. I have applied one of them by doing waste management. I made a special channel to dispose of cracker waste so that it does not release odor nor pollute the surrounding environment." |
|---|---|
| Company 2 | "I understand how difficult it is for plastic packaging to decompose, which is detrimental to our environment. Hence, I use environmentally friendly materials for my product packaging." |
| Company 4 | "Two years ago, I tried to use wood charcoal briquettes as fuel in our factory. Back in university, I learned about safe fuel selection, so I tried to apply it to my business. The results were satisfying; I could save more by using briquettes as fuel in the tofu production process." |

## 5. Discussion

This study found similarities across Sundanese small industries such that, in general, they could survive across generations because the strength of traditional Sundanese cultural traditions is translated through the intrinsic values of entrepreneurs into their business activities. In addition, there are several other important factors that determine longevity [11–13]. In the Sundanese ethnicity, the tradition of high long-term orientation, highcollectivism, low power distance, and high indulgence positively contribute to the longevity of small industry in Indonesia. This makes sense, because Indonesia as a country

is characterized by the strength of collective culture, social construction and the strength of traditional values that are always passed down from generation to generation [44]. Therefore, the local values applied by Sundanese entrepreneurs that are translated into their business practices become their strength in running their businesses so that they last for at least three generations. However, any proposition arguing that the dimension of power distance tends to contradict flexibility is not fully proven [14]. This study shows that the freedom granted by the government by providing flexibility for small industries to develop, by not binding them with rigid and burdensome rules, makes small industries more flexible in terms of entrepreneurship. "Freedom" is essential for entrepreneurs to run their business activities and to be creative [45].

On the other hand, some traditional Sundanese cultural values negatively influence their entrepreneurial values, namely high uncertainty avoidance and high masculinity, which actually hinder small industries from growing. The low risk taking of Sundanese entrepreneurs has an impact on their low level of innovation, whereas other studies in Indonesia show that there is a positive impact of risk-taking behaviour on the performance of small industries [46]. High innovation will encourage small industries to be more competitive in the midst of competition and have a positive influence on their economic performance [47]. The same thing was also argued by [48], who revealed that innovation is the key to the success of small industries, in that it needs to be supported by factors such as management skills, technological capacity, financial factors, and the size of the company itself. Researchers also saw innovation as one of the key competitive advantages organizations must possess in the 21st century [49]. In particular, small industries should find strategies to improve their ability to innovate effectively [50] because it is the key to success for small players in a competitive global environment. At the same time, innovative and competitive small industries are positively related to future growth in developing countries [51,52]. This study is in line with [53], saying that uncertainty avoidance negatively affects risk-taking. Therefore, Sundanese entrepreneurs must have good and calculated risk-taking to be more innovative in facing challenges and competition to achieve better economic performance. They must embrace global culture, namely technological innovation and digitalization, that helps improve performance. Thus, it is hoped that critical factors that influence economic performance, such as sales growth, market share, cost reduction, and profitability, will also experience significant improvement.

Another Sundanese cultural tradition to consider in achieving better economic performance is masculinity. The Sundanese ethnicity emphasizes the power of men in many life aspects, including business. Masculine culture in the Sundanese tends to lead to an ambitious and individualistic attitude, avoiding the acceptance of new things or new people from outside the community. The high culture of masculinity also affects the achievement of their goals. This finding is in line with research conducted by Graham [54]. In the Sundanese, for example, many male entrepreneurs symbolize the success of their business by prioritizing ownership of vehicles, houses, land, and other symbols of luxury rather than using their profits to improve company capacity, which would boost the economic performance. This factor leads to their business performance being economically undeveloped. Therefore, the dimensions of masculinity vs. femininity should be balanced [55,56].

Concerning the social and environmental performance in small industries, our findings are in line with research conducted by Famiola [44], showing that the strong values of entrepreneurs are important factors for creating socially and environmentally conscious businesses in small industries. The social motives of Sundanese entrepreneurs are based on their religious values and highly collectivist culture. In addition, the formal education experience of entrepreneurs significantly affects their environmental interests.

*Limitations and Future Research Agenda*

The main limitation of this study is the relatively small sample size because of the limited number of small industries that survived three generations in Indonesia. This finding only investigated Sundanese small industries, and thus it cannot be generalized to

others. However, the findings can be used as a basis for other ethnic groups that may have similar characteristics. Therefore, future research should test a hypothetical proposition through a survey to produce a more robust conclusion.

## 6. Conclusions

This research shows that traditional culture can be a source of strength for small industries to survive across generations, but that it can also hinder their business performance. Sundanese cultural traditions which include high long-term orientation, highcollectivism, low power distance, and high indulgence, reflected through the strong values of entrepreneurs in their business activities, are important factors that enable small industries to endure. However, Sundanese cultural traditions, especially the high uncertainty avoidance and high masculinity, negatively affect their economic performance. The social motives of Sundanese entrepreneurs are based on their religious values and highly collectivist culture. In addition, the formal education experience of entrepreneurs significantly affects their environmental interests. This study found that Hofstede's cultural dimension plays less of a role in the context of environmental performance in small industries. At the same time, this research also reveals that companies' ability to endure (longevity) does not guarantee that they can be sustainable.

We recommend efforts to change the mindset of Sundanese entrepreneurs to be more open to an innovative global culture that can support their performance development and free themselves from the hereditary culture hindering their businesses. Combining a positive traditional culture with an innovative global culture and a focus on improving the critical factors affecting economic performance will help Sundanese small industries achieve sustainability. Sustainable entrepreneurship education must be considered carefully. It can be designed in the higher education curriculum to produce graduates who are not only capable of entrepreneurship but who also have a balanced mindset between economic, social, and environmental objectives as the core values of business. The right education ecosystem to support a sustainable entrepreneurship agenda should be carefully developed. The government's role in linking small industries with business associations or networks will also improve the business performance of small industries. Another practical implication of this research is the importance of support from all parties to build sustainable entrepreneurship in small industries in Indonesia while maintaining the uniqueness of small industries and traditional values that contribute positively to business, combined with an innovative global culture that supports the performance of small industries in Indonesia. Hence, small industries that both survive over the long-term and that are economically, socially, and environmentally sustainable are realized.

**Author Contributions:** Conceptualization, A.C. and G.K.; Data curation, A.C., G.K. and A.M.; Formal analysis, A.C. and K.M.; Funding acquisition, A.C. and A.M.; Investigation, A.C. and G.K.; Methodology, A.C. and K.M.; Project administration, A.C., A.M. and K.M.; Resources, A.C.; Software, A.C.; Supervision, A.C. and G.K.; Validation, G.K., A.M. and K.M.; Visualization, A.C.; Writing— original draft, A.C. and G.K.; Writing—review & editing, A.C. and A.M. All authors have read and agreed to the published version of the manuscript.

**Funding:** This research received no external funding. And "The APC was funded by Universitas Padjadjaran".

**Informed Consent Statement:** Informed consent was obtained from all subjects involved in the study.

**Data Availability Statement:** The study did not publish any data online.

**Acknowledgments:** We would like to extend our thanks to all participating companies, which took the time to speak with us. Without their help and assistance, the outcome of the paper would not have been feasible.

**Conflicts of Interest:** The authors declare that they have no conflict of interest.

## Appendix A

**Table A1.** The Characteristics of the Small Industries Analyzed.

| Company | Type of Small Industry | Product Description | Year of Establishment | Location | Number of Employees | Turnover/Year (IDR) * |
|---|---|---|---|---|---|---|
| 1 | Cracker Industry | It only produces one product, namely *aci* crackers. Crackers are a traditional Sundanese food made of tapioca flour mixed with fish flavorings. Crackers are made by steaming the dough until it is cooked, then cut into thin strips, dried in the sun, and deep fried. | 1961 | Ciamis District | 14 | IDR.280,000,000. |
| 2 | *Dodol* Industry | It produces one product, namely *dodol*, with various flavors (original, sesame, and fruits). *Dodol* is a typical traditional food of Garut. It is a sticky, sweet cake made of glutinous rice flour, palm sugar, coconut milk, and salt. | 1964 | Garut District | 11 | IDR.210,000,000. |
| 3 | Handycraft Industry | It produces typical Tasikmalaya handicrafts made of pandanus or rattan woven with certain motifs. The products are usually bags, wallets, kitchen utensils, and others. | 1960 | Tasikmalaya District | 10 | IDR.200,000,000. |
| 4 | Tofu Industry | It only produces one product: fried tofu. Tofu is made of the coagulation of soybean juice. Tofu is a typical food from Sumedang District. | 1971 | Sumedang District | 14 | IDR.285,000,000. |
| 5 | *Pisang Sale* Industry | It only produces one: *pisang sale* (sweetened banana chips), which is processed food made of bananas that are cut into thin slices and dried in the sun. The purpose of drying is to reduce the water content of the bananas to last longer. It can be eaten directly or fried with flour first. | 1985 | Ciamis District | 11 | IDR.195,000,000. |
| 6 | Cassava Chips Industry | It only produces one product: cassava chips. | 1981 | Ciamis District | 10 | IDR.180,000,000. |
| 7 | Copra Industry | It only produces one product: dried copra, the dried flesh of the coconut. Copra is one of the most important coconut derivative products, the raw material for making coconut oil and its derivatives. | 1972 | Banjar District | 11 | IDR.290,000,000. |
| 8 | *Opak* Industry | It only produces one product: *opak*, a typical Sundanese snack. It is dry and crunchy, similar to crackers. *Opak* is made of rice or glutinous rice flour seasoned with salt, sugar, grated coconut, and seasonings. | 1985 | Ciamis District | 10 | IDR.175,000,000. |
| 9 | Banana Chips Industry | It only produces one product: banana chips, which are made of bananas that are thinly sliced and deep fried. | 1982 | Ciamis District | 10 | IDR.185,000,000. |
| 10 | *Rengginang* Industry | It only produces one product: *Rengginang*, a traditional Sundanese food made of dried glutinous rice. It can be salty and sweet. The dough is round, dried in the sun, and deep-fried. | 1980 | Tasikmalaya District | 11 | IDR.200,000,000. |

Source: The results of Preliminary Study (2019). Notes: * 1 US$ in 2019–2020 = IDR.14,050.-.

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
