# Peer review of "The Impacts of Traditional Culture on Small Industries Longevity and Sustainability: A Case on Sundanese in Indonesia"

_sustainability, doi:10.3390/su142114445_

Round 1

Reviewer 1 Report

Thank you for the opportunity to review the article “The Impact of Traditional Culture for Longevity and Small Industries Sustainability: Case Study on The Sundanese Ethnic Group in Indonesia”. The paper addresses an interesting and well researched theme in the recent period about exploring the extent to which the cultural dimension is able to affect the performance of small industries in achieving sustainability.

This study represents a solid effort in the field approached. It is constructed in a mature manner, following the publication standards of the journal, discussing the subject that needs to be comprehensively analyzed, “by understanding the characteristics and traditions of Sundanese culture, uniqueness, and learning the limitations of small industries belonging to the Sundanese ethnic group, they will be able to help them achieve business sustainability”, as the authors underline.

Also, the study is written in an adequate manner for the reader, with a specific review of the literature and robust research design. The results are presented clearly and coherently, using visuals and text. The tables constructed in the paper were relevant to explore the results of the research and the ways these were adapted to the explanations in the text.

As the authors say, one conclusion of this paper assumes that “the longevity capability in small industries does not guarantee that the business can be categorized as sustainable”.

Moreover, there are some point-by-point observations that should be addressed in this revision.

-          In the title it is found the idea of “case study” but in text the reader finds out about multiple case study and several instruments typical of qualitative methods, namely documentary style analysis and direct interviews, etc. I propose to be clear about the content when using words in the title and make the changes that are required

-          The Abstract is very poorly written in terms of English language and past tense, very different in style from the rest of the text. Please modify the Abstract to be in line and the logic of the article

-          Reformulate Line 88-89 “Hofstede's theory of cultural dimensions is a framework for cross-cultural communication, developed by Geert Hofstede”

-          Reformulate Line 115 “we begin the discussion by explaining the introduction”

-          Better formulation of the sections of the article from Line 113-118.

-          Please add very clear for the reader what are the objectives of the study.

-          The conclusion from Line 549-550 is a commonsense idea “we suggest the need for efforts to change the mindset of Sundanese entrepreneurs to be more open to an innovative global culture”, please make sure to be in line with the objectives of the research.

-          Please correct all the typos in the text and make correct use of capital letters in English where they supposed to be present and vice-versa.

Author Response

Dear reviewer, 

My response regarding your suggestion, I include it in the file bellow

Thank you

Reviewer 2 Report

I would like to thank the authors and the journal editor for the oportunity to review this manuscript. The idea to investigate small business sustainability from the point of Hofstede's cultural dimensions is interesting and still relevant. I generally commend the authors for their detailed work and a story that they depicted so vividly. I will point to a few smaller issues that may be constructive for the final version of this manuscript.

- In the title the phrase "case study" is not appropriate, since there was more than one subject observed and because the approach is generally not a typical case study - please consult literature on how to name your research approach.

- The 1. Introduction part should be more concise and more focused on the problem you are trying to solve. I would like to see this section shortened to about 50% of its current lenght, eliminating the parts and facts that are not central to your issue. Also, in this part you need a clearer definition of your research problem, an argumentation on why you used Hofstede's Cultural Theory, an explanation on why you used your research method the way you did. 

- Part 2. Literature is written well, I have no issues here. 

- Part 3. Methods lacks, before part 3.1, a short introduction in which a rationalle for the methods used is presented. The authors need to explain why they opted for the type of research which they performed. I am not convinced that the authors used multiple case study as their main research approach - the questions used and the subsequent results presentation do not resemble case studies. The authors should find a better name for their research method, as the cited references [29] and [30] are not favourable for the "multiple case study" you have presented. This is the biggest issue of this manuscript, and should be addressed.

- Authors should explain how did they choose the 10 observed cases, and why did they approach 13 cases in total. Please identify what type of sample this is. What was the background of contacting only these cases? How many companies like this were in this area at all? It is not acceptable to present the observed cases without at least information about number of employees, number of products, reported turnover. 

- Authors should also provide information on who conducted the interviews, how long did the interviews last, how were the responses recorded?

- The authors name Table 2 as a semi-structured questionnaire, but this is not true, as a simple three point scales are presented for every item. This needs to be properly named according to the literature on research methodology.

- Part 3.2 Data Analysis should provide more information on the steps you have taken. There should be enough information on your procedures to ensure replication of your study by other researchers. 

- Parts 4. and 5. are well written and provide enderstandable information.

- Part 6. Conclusion lacks mentioning study limitations.

- The manuscript should be checked for language and typing errors and consistency.

Author Response

Dear Reviewer,

My response regarding your suggestion, I include it in the file bellow

Thank you

Round 2

Reviewer 1 Report

Thank you for the opportunity to review the paper “The Impact of Traditional Culture for Longevity and Small Industries Sustainability: Case Study on The Sundanese Ethnic Group in Indonesia”.   

The authors responded/ explained with reasonable arguments and corrected all the remarks and observations highlighted in the previous review and the results suggest a more consistent and logical text with a clearer reference list.

To sum it up, the authors developed a more in-depths theoretical presentation about the subject, integrating the suggested aspects of the review, a rewritten Abstract, main objectives and the conclusion are also modified, and the reference list is updated.

As minor observations, I suggest to:

1.      Readjust the Figure 1 and Figure 2, as they are very poor quality, maybe from the pdf conversion, but still these are not publishable figures

2.      Correct Table 6 for that long vertical text, Table 9, Table 10 to be centered as the others

3.      Table 1 should be defined as an Appendix, at the end of the paper

I consider that the paper is publishable with minor revisions and after a final check from the authors for some typing errors.

Author Response

Dear Reviewers,

Thank you for your advice.

My response is attached in the file below

Thank You
